# Understanding Scaling Laws With Token-Level Analysis

**Arkil Patel, Marius Mosbach**
Mila and McGill University
{arkil.patel,marius.mosbach}@mila.quebec

**Siva Reddy**
Mila and McGill University
Canada CIFAR AI Chair
ServiceNow Research
siva.reddy@mila.quebec

**Dzmitry Bahdanau**
Mila and McGill University
Canada CIFAR AI Chair
Periodic Labs
bahdanau@mila.quebec

## Abstract

Large language models (LLMs) exhibit remarkably smooth *power-law* scaling of cross-entropy loss with respect to training data size (tokens). Despite the robustness of this empirical law, its mechanistic origins remain unclear: why do the dynamics of gradient-based training yield such a clean functional form? This paper attacks the question through a *token-level* lens. We pretrain OLMo-2 across multiple token budgets and log per-token validation losses across checkpoints. We find (i) individual token-loss trajectories are highly heterogeneous and often noisy, with no obvious shared parametric form; (ii) nevertheless, best-fit power laws typically imply decreasing trends, and a substantial fraction of tokens are well-approximated by power laws; and (iii) the canonical aggregate power law emerges only after averaging over a critical mass of token-level losses, with fit error decaying rapidly as coverage increases. Our findings show that power-law scaling is not a universal property of each token's learning curve, but an emergent phenomenon arising from averaging many heterogeneous token-wise dynamics.

## 1 Introduction

Power-law scaling relationships are one of the most practically useful and scientifically intriguing regularities in modern deep learning. For autoregressive language modeling, validation loss often follows a law of the form

$$\mathcal{L}(D) \approx C + A\,D^{-\alpha}, \tag{1}$$

over multiple orders of magnitude in training data D (Kaplan et al., 2020; Hoffmann et al., 2022) as illustrated in Figure 1 *Left*. Such *scaling laws* enable forecasting, compute/data allocation, and model-selection strategies. However, the *origin* of the smooth power law is still not well-understood. This work focuses on a single guiding research question: *Why do LLMs exhibit power-law scaling with respect to the number of training tokens?*

We approach this question by decomposing aggregate behavior into *token-level* learning dynamics. Each token occurrence in a fixed validation set defines a *trajectory*: its cross-entropy loss as a function of training tokens consumed. In experiments pretraining a small OLMo language model (Team et al., 2024), we find that token-level trajectories are noisy (Figure 1 *Right*) and do not share any common parametric form that could directly explain the aggregate power law.

Nevertheless, when fit individually, a substantial fraction of token-level trajectories are well-approximated by *their own* power laws (Figure 2 *Left*), typically implying a net decreasing trend with training. However, these token-wise fits are heterogeneous in their parameters and do not trivially "average" into the aggregate law. To relate the smooth aggregate curve to this micro-level diversity, we run *coverage* experiments in which we average loss over random subsets of tokens and fit a power law to the resulting subset-mean trajectory. As coverage grows beyond a *critical mass of tokens*, fit

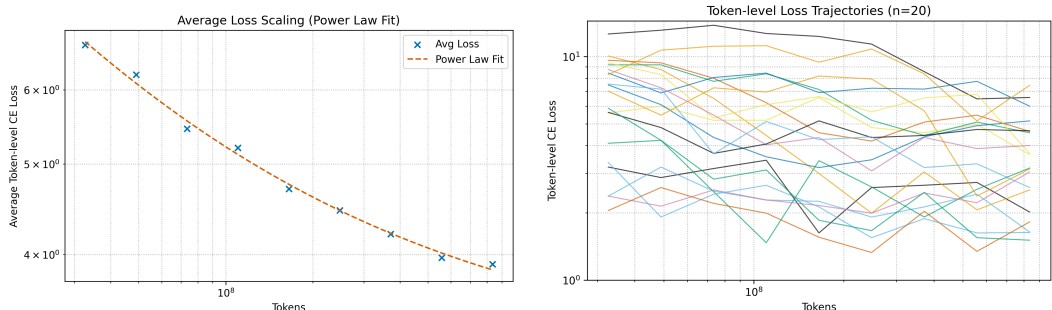

Figure 1: *Left:* **Aggregate loss follows a clean power law.** Average token-level validation cross-entropy versus training tokens for OLMo-2 (20M), with a best-fit power-law curve (Equation 1) over the observed budgets. This is the phenomenon we aim to explain. *Right:* **Random token-level loss trajectories are noisy and diverse.** Each curve shows a randomly sampled token's validation cross-entropy as a function of training tokens. Trajectories vary widely in scale and often display non-monotonicity across checkpoints, suggesting there is no single canonical functional form.

quality improves sharply (Figures 2 *Right* and 3), and subset-average trajectories converge toward the same global trend measured on the full validation set (Appendix Figure 7).

Finally, we test whether token-level noisiness on real text is partly an artifact of using a one-sample estimate of cross-entropy, since the ground-truth next-token distribution is unknown. We run a controlled synthetic experiment where sequences are generated from a known trigram distribution, enabling computation of the *true* cross-entropy. We find similar results: the validation-set average exhibits smooth power-law scaling, while token-level trajectories remain diverse and irregular across budgets (Appendix Figures 4 and 8).

**Background and Related Work.** Empirical power-law scaling laws have been well-documented across tasks, architectures, and modalities (Hestness et al., 2017; Henighan et al., 2020; Kaplan et al., 2020; Hoffmann et al., 2022). Several works aim to explain the origin of this behavior. Bahri et al. (2024) formalize two regimes, variance-limited and resolution-limited, linking power-law exponents to spectral properties of kernels under data manifold assumptions. Michaud et al. (2023) propose the Quantization Model, which attributes scaling to learning discrete "quanta" of capabilities ordered by difficulty and prevalence. Havrilla & Liao (2024) derive power-law generalization bounds by assuming that training data lie on a low-dimensional manifold. In contrast to these works, we analyze scaling laws by decomposing aggregate loss into token-level trajectories. Our findings are empirical and do not make any strong assumptions about smoothness or low intrinsic dimension.

## 2 EXPERIMENTAL SETUP

**Models and training.** We train OLMo-2 (Team et al., 2024) at 20M parameters (with additional 150M runs showing similar qualitative behavior as seen in Appendix C). For each configuration, we run multiple training budgets, measured in total tokens processed. At the end of each run, we evaluate on a fixed validation set $\mathcal{V}$ and record *per-token* cross-entropy losses.

**Token-level trajectories.** Let the validation set contain $|\mathcal{V}|$ tokens[1], indexed by $i \in \{1, \dots, |\mathcal{V}|\}$. For a given training budget D, we define the token-level loss

$$\ell_i(D) = -\log p_\theta(x_i \mid x_{<i}), \tag{2}$$

where $\theta$ denotes the model parameters. The aggregate loss is the mean

$$\mathcal{L}(D) = \frac{1}{|\mathcal{V}|} \sum_{i=1}^{|\mathcal{V}|} \ell_i(D). \tag{3}$$

---

[1]By "token", we mean a *contextual token*: each token occurrence in the validation set together with its context (position and preceding tokens). Equivalently, $|\mathcal{V}|$ counts token occurrences (unique contextualized instances), not the distinct token identities in the vocabulary.

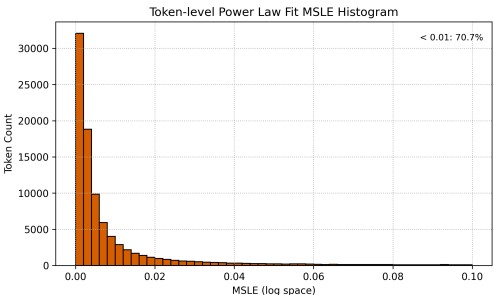 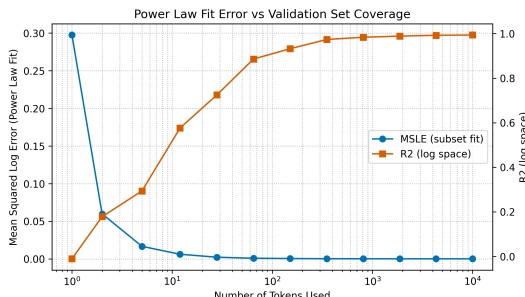

Figure 2: *Left*: **Many token-level trajectories admit low-error power-law fits.** Histogram of per-token MSLE between observed losses and their best-fit power laws (log space). *Right*: **Power-law fit quality improves with token coverage.** MSLE (left axis) and $R^2$ (right axis) for power-law fits under subset averaging: for each $|S|$ (x-axis), we draw 20 random subsets of $|S|$ tokens; within each subset we compute the mean loss at every token budget, fit a power law to this mean-loss curve, and then report MSLE and $R^2$ averaged across the $m$ fits. We see a sharp improvement as $|S|$ increases, indicating that averaging over a *critical mass* of tokens is required to observe clean scaling behaviour.

**Aggregate power-law fit.** We fit a power law to the *aggregate* validation loss

$$\mathcal{L}(D) = C + A\,D^{-B}, \tag{4}$$

where $C$ is an irreducible loss floor, $A > 0$ is a scale parameter, and $B > 0$ is the data-scaling exponent. Figure 1 *Left* shows that the resulting fit tracks the observed average loss closely over the range of token budgets studied. This reproduces the qualitative phenomenon reported in prior work on data scaling in language modeling (Kaplan et al., 2020; Hoffmann et al., 2022).

**Per-token power-law fits.** For each token $i$, we fit a 3-parameter power law in *log space*:

$$\ell_i(D) \approx c_i + a_i\,D^{-b_i}, \tag{5}$$

where $c_i \geq 0$ and $a_i, b_i > 0$. We quantify fit quality using mean squared log error (MSLE) computed on the set of available budgets for token $i$.

**Coverage / critical mass experiments.** To study how power-law behavior emerges with averaging, we form *partial averages* over subsets $S \subseteq \{1, \ldots, |\mathcal{V}|\}$:

$$\mathcal{L}_S(D) = \frac{1}{|S|} \sum_{i \in S} \ell_i(D). \tag{6}$$

We vary $|S|$ ("coverage") and fit Equation 4 to $\mathcal{L}_S(D)$. We report (i) the MSLE of the subset-fit itself and (ii) error with respect to the *full-set* best-fit power law (fit on $S = \mathcal{V}$).

**Synthetic trigram experiments.** To separate learning-curve heterogeneity from one-sample cross-entropy estimation noise, we train a GPT-2 style transformer (150M parameters) on synthetic sequences generated by a trigram process with known conditionals. We provide data generation details in Appendix A. Because the ground-truth next-token distribution $p^\star(\cdot \mid h)$ is known for every trigram context $h = (x_{t-2}, x_{t-1})$, we can compute the *true* cross-entropy at each validation position:

$$\ell_i^{\text{true}}(D) = -\sum_{w \in \mathcal{V}} p^\star(w \mid h_i) \log p_\theta(w \mid h_i), \tag{7}$$

where $h_i$ denotes the length-2 context at position $i$.

## 3 RESULTS AND DISCUSSION

**Token-level trajectories are heterogeneous and noisy.** One naive hypothesis is that token-level curves share a common shape and that the aggregate power law simply reflects most tokens following Equation 5. However, this doesn't seem to be true and instead we observe substantial idiosyncrasy as seen in Figure 1 *Right*, which plots 20 randomly sampled token trajectories. We see that many curves are non-monotone over budgets, and several exhibit sharp fluctuations.

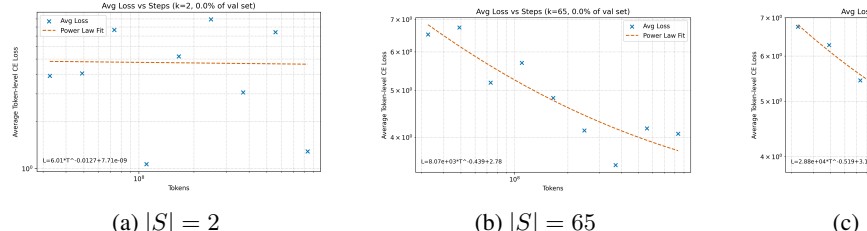

(a) $|S| = 2$        (b) $|S| = 65$        (c) $|S| = 4328$

Figure 3: **Power laws become visually cleaner as we average more tokens.** Each panel plots the mean validation loss over a random subset of $|S|$ tokens, across training budgets, with a best-fit power law. The curve transitions from noisy (small $|S|$) to smooth and tightly fit by a power law (large $|S|$).

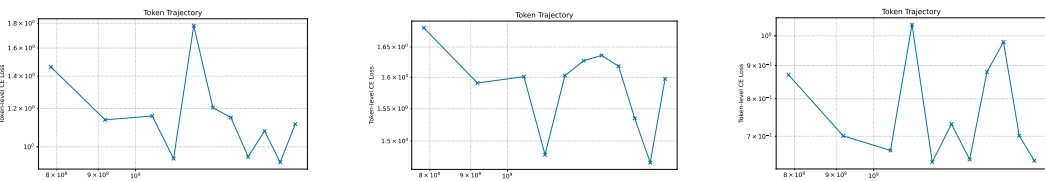

Figure 4: **Token-level *true* cross-entropy trajectories for the trigram LM.** Each panel shows a different randomly selected validation token's true cross-entropy. Despite eliminating one-sample estimation noise, trajectories remain noisy and heterogeneous, mirroring the natural-text setting.

**A majority of tokens are well-approximated by power laws.** Although Figure 1 *Right* highlights substantial irregularity at the token level, many tokens are nonetheless well-captured by a simple power-law model when fit individually in log space. Figure 2 *Left* summarizes this phenomenon quantitatively: the distribution of per-token MSLE exhibits a pronounced mass near zero, indicating that a majority of tokens have trajectories that are close to their best-fit power laws across the observed token budgets. For instance, we see that around 70% of the token-level trajectories exhibit an MSLE $< 0.01$. To illustrate this further, Figure 5 in the Appendix shows some random selections of such *low-MSLE* token trajectories. Moreover, we find that the best-fit power law for each token trajectory, irrespective of quality of fit, always indicates a decreasing trend (examples in Appendix Figure 6).

**Aggregate Power-law emerges on averaging over a critical mass of tokens.** The most striking result is that *fit quality improves rapidly with coverage*. As we increase $|S|$, the subset-average $\mathcal{L}_S(D)$ becomes smoother and more power-law-like. Figure 2 *Right* quantifies this effect: MSLE drops steeply and $R^2$ in log space rises toward $\approx 1$ as $|S|$ grows. Crucially, the same trend appears whether we evaluate error against the subset's own best-fit power law or against the *full validation-set* best-fit curve (Appendix Figure 7), indicating that subset averages not only become power-law-like, but also converge toward the same global scaling trend. We can also see the transition visually. Figure 3 shows average-loss trajectories for increasing $|S|$. At small $|S|$, the curve is jagged and deviates from a smooth law. As $|S|$ grows into the hundreds, the power law becomes visually "clean" and stable.

**The same phenomenon holds under *true* cross-entropy (synthetic trigram control).** A natural concern is that token-level noisiness in Figure 1 *Right* could be driven by the one-sample nature of cross-entropy estimates on real text. In a synthetic trigram setting, we can evaluate the model under the *true* cross-entropy induced by the known data-generating distribution. Nonetheless, we again observe smooth power-law scaling for the validation-set average (Appendix Figure 8) alongside noisy, heterogeneous token-level trajectories (Figure 4).

## 4 CONCLUSION

We presented a token-level study of data scaling in OLMo-2 language models. Token-wise loss trajectories are heterogeneous and frequently noisy, yet many tokens admit low-error power-law fits. Most importantly, the canonical smooth power law in aggregate loss *emerges only after averaging over a critical mass of tokens*, with fit error rapidly decreasing as coverage increases.

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

## A    TRIGRAM DATA GENERATION AND TRUE CROSS-ENTROPY COMPUTATION

We generate synthetic pretraining and validation data from a trigram process over a finite vocabulary $\mathcal{V} = \{0, 1, \ldots, V - 1\}$. Each position $t$ has context $h_t = (x_{t-2}, x_{t-1})$, so the context space is $\mathcal{H} = \mathcal{V} \times \mathcal{V}$ with $|\mathcal{H}| = V^2$. For each context $h \in \mathcal{H}$, we sample a scalar concentration $\alpha_h$ and then draw a next-token distribution $p^\star(\cdot \mid h)$ from a *symmetric Dirichlet*:

$$p^\star(\cdot \mid h) \sim \text{Dirichlet}(\underbrace{\alpha_h, \ldots, \alpha_h}_{V \text{ times}}). \tag{8}$$

Specifically, we sample i.i.d. Gamma variables $g_{h,w} \sim \text{Gamma}(\alpha_h, 1)$ for each $w \in \mathcal{V}$ and normalize:

$$p^\star(w \mid h) = \frac{g_{h,w}}{\sum_{u \in \mathcal{V}} g_{h,u}}. \tag{9}$$

The scalar $\alpha_h$ controls how peaked the distribution is: lower $\alpha_h < 1$ yields sparse (spiky) rows, while higher $\alpha_h \gg 1$ yields near-uniform rows.

Collecting all rows yields a probability matrix $P \in \mathbb{R}^{V^2 \times V}$, where row $P[h]$ corresponds to $p^\star(\cdot \mid h)$.

To sample a length-$T$ sequence $\{x_1, \ldots, x_T\}$, we draw $x_1, x_2 \sim \text{Uniform}(\mathcal{V})$ and then iterate for $t = 3, \ldots, T$:

$$h_t = (x_{t-2}, x_{t-1}), \qquad x_t \sim p^\star(\cdot \mid h_t). \tag{10}$$

Unlike natural text, the ground-truth conditional $p^\star(\cdot \mid h)$ is known, enabling exact computation of the *true cross-entropy* at each validation position $i$:

$$\ell_i^{\text{true}}(D) = -\sum_{w \in \mathcal{V}} p^\star(w \mid h_i) \log p_\theta(w \mid h_i). \tag{11}$$

This removes one-sample estimation noise and allows a controlled comparison to the real-corpus setting.

## B    EXPERIMENTAL DETAILS

We make use of the training codebase for OLMo (Team et al., 2024).[2] We follow the model configuration and parameters used in DataDecide's experiments (Magnusson et al., 2025) (see Table 2 in their paper) and pretrained on varying token budgets of the Dolma dataset (Soldaini et al., 2024). For the trigram LM experiments, we implemented a GPT-2 LM following the codebase of nanoGPT (Karpathy, 2023).[3] All pretraining was carried out over a set of 4 H100 GPUs.

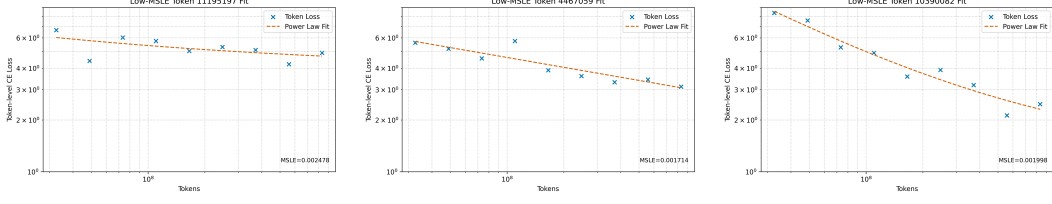

(a) Low-MSLE token example 1.    (b) Low-MSLE token example 2.    (c) Low-MSLE token example 3.

Figure 5: **Examples of individual token trajectories with good power-law fits.** Each panel shows a token's validation loss across training token budgets (markers) along with its best-fit power law (dashed). Tokens are selected randomly from the low-MSLE subset (left tail of Figure 2 *Left*: MSLE < 0.01). We observe that strong power-law behavior occurs at the individual-token level for a substantial fraction of tokens.

---

[2]https://github.com/allenai/OLMo
[3]https://github.com/karpathy/nanoGPT

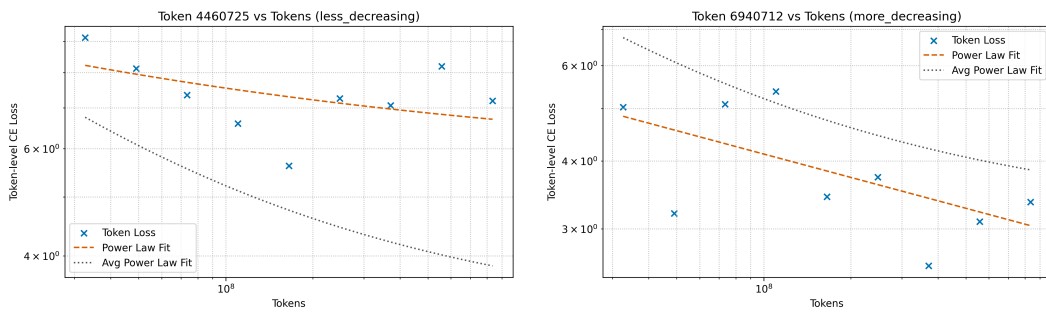

(a) A weakly decreasing token trajectory.          (b) A strongly decreasing token trajectory.

Figure 6: **Example token-wise power-law fits show decreasing loss trends.** Blue markers are token losses at each budget, dashed lines are best-fit token-wise power laws (Equation 5), and the dotted curve shows the aggregate-fit power law for comparison.

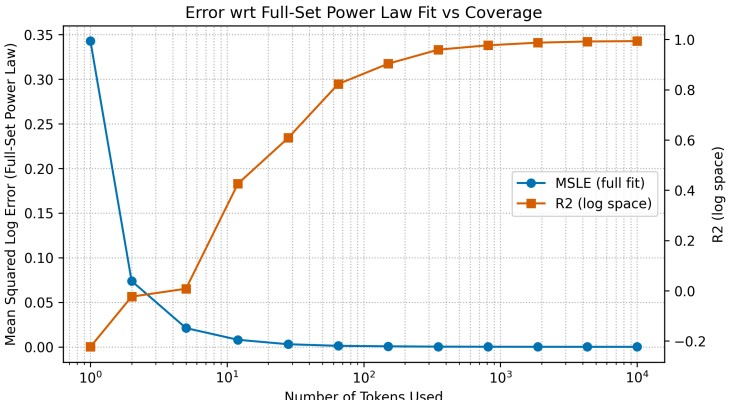

Figure 7: **Subset averages converge toward the full-set power-law fit.** We evaluate MSLE (left axis) and $R^2$ (right axis) between each subset-averaged loss curve and the global best-fit power law. Results are averaged over $m = 20$ random subsets per subset size $|S|$ (x-axis). Error drops rapidly with coverage, indicating that partial averages align with the full-set trend after aggregating over a *critical mass* of tokens.

## C   ADDITIONAL RESULTS

Figure 5 shows representative token trajectories randomly sampled from the low-MSLE subset (MSLE $< 0.01$). These examples indicate that a substantial number of tokens follow relatively clean, decreasing power-law trends, even though the overall population includes many irregular cases. Notably, token-wise fits often diverge from the aggregate-fit slope, underscoring that the global exponent does not arise from a shared parametric form across tokens.

Figure 6 shows two representative token-level trajectories, one weakly decreasing and one strongly decreasing, based on how their loss trends compare to the aggregate. Despite differing in slope and fit quality, both best-fit power laws exhibit a consistent decreasing trend, illustrating that fitted exponents are typically positive even for noisy curves.

Figure 7 quantifies how subset-averaged loss trajectories approach the global trend. Even when fitting a power law to a small subset $S$, we observe that the mean loss trajectory becomes increasingly similar to the full validation-set fit as $|S|$ increases.

Figure 8 shows that even when computing exact cross-entropy under the known data-generating distribution, the same qualitative trends emerge: aggregate loss follows a smooth power law, while individual token-level curves remain noisy and diverse.

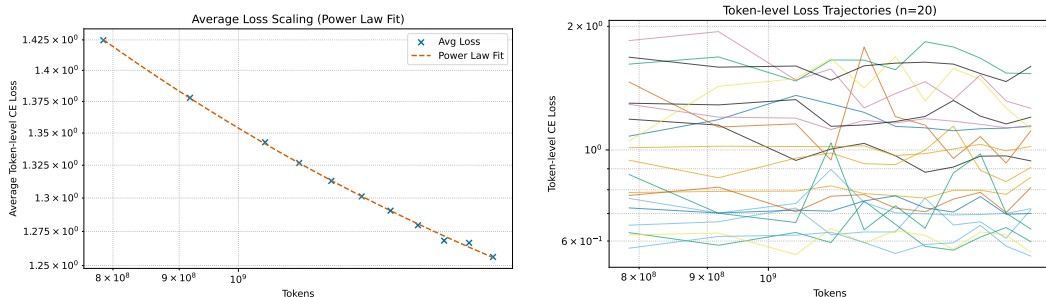

Figure 8: **True cross-entropy analysis on synthetic trigram data.** *Left*: Average validation loss computed using the known ground-truth distribution follows a clean power law. *Right*: Individual token-level losses remain noisy and heterogeneous across training budgets. These results mirror the natural-text setting, indicating that our findings are not artifacts of estimation noise.

Figures 9 and 10 show results for a 150M parameter OLMo model. The same trends persist: token-level trajectories are heterogeneous and often noisy, but fit quality improves rapidly with coverage, and the aggregate loss follows a clean power law.

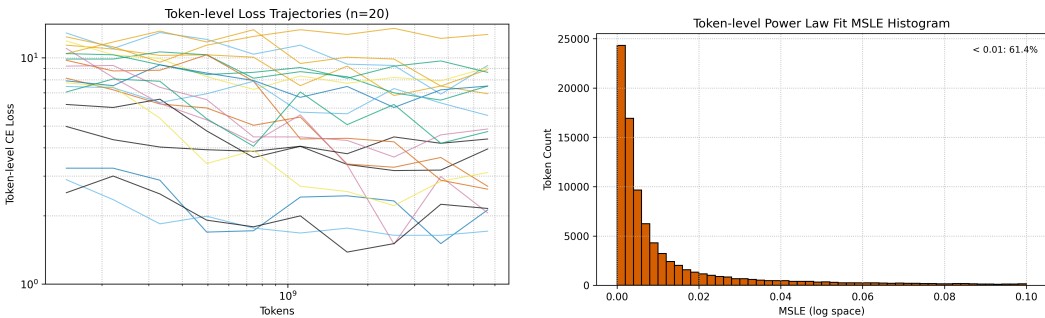

Figure 9: **Token-level heterogeneity in the 150M model.** *Left*: Random token trajectories remain noisy and non-monotonic. *Right*: A substantial fraction of tokens are well-fit by individual power laws (low MSLE), similar to the 20M model.

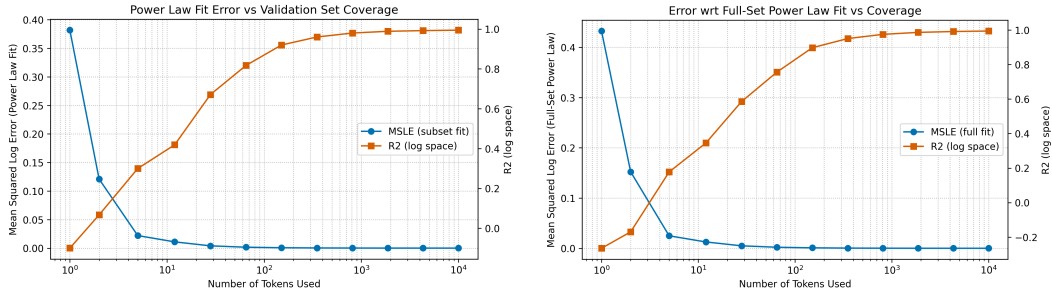

Figure 10: **Coverage analysis for the 150M model.** *Left*: Power-law fit quality improves rapidly with subset size. *Right*: Subset-averaged loss curves converge to the full-set fit, reinforcing that clean scaling behavior emerges through aggregation.

