# OpenReview forum: "Understanding Scaling Laws With Token-Level Analysis"
_ICLR.cc/2026/Workshop/Sci4DL — Sci4DL 2026_

### Official Review · Reviewer_onGu · 2026-02-24

**Fit:** 3
**Significance:** 2
**Confidence:** 3

**Summary:**

In this paper authors investigate a phenomenon of neural scaling laws through the lens of individual tokens. They found out that token-level trajectories are highly heterogenous and noisy and the canonical power-law emerges only after averaging across a critical mass of token-level losses. Moreover,  even though all token-level trajectories look noisy, the majority of them still exhibit some kind of power law behavior (MSLE < 0.01 for the best fit). The same phenomenon holds true even for synthetic experiments with trigram LMs, where the underlying cross-entropy is known.

**Strengths:**

- The explanation of scaling laws for language models is an important topic and token-level trajectories add more insight in our understanding of the emergence of power laws.
- Paper is well-written and all concepts are laid out concisely.

**Suggestions:**

- What is the validation set V in the experiments with OLMo? Just a part Dolma?
- Is there a connection between the majority of token losses exhibiting power law, the distribution of tokens in the training/validation dataset and the emergence of the power law when increasing coverage? How the distribution the token-level power law fit histogram would look like for trigram LMs?

---

### Official Review · Reviewer_xxSr · 2026-02-25

**Fit:** 1
**Significance:** 1
**Confidence:** 2

**Summary:**

The paper examines whether the commonly observed power-law scaling of validation cross-entropy with training token budget is reflected at the level of individual validation tokens. It finds that per-token loss trajectories are noisy, though generally decreasing with more training data. A smooth power-law relationship emerges reliably only after averaging loss trajectories over a sufficiently large set of tokens.

**Strengths:**

Logging per-token statistics is a potentially a valuable direction. The paper performs experiments at a reasonable scale towards this goal.

**Suggestions:**

Power-law fits become cleaner as you average more tokens can largely be explained by variance reduction from averaging, independent of anything specifically power-law about learning dynamics. More substantive question is whether the token-level behavior has systematic structure - especially dependence on context (e.g., position, preceding tokens, semantic features) and whether that structure explains the aggregate scaling or the location of the critical mass.

---

### Official Review · Reviewer_SMsE · 2026-02-27

**Fit:** 3
**Significance:** 2
**Confidence:** 1

**Summary:**

This paper aims to understand the neural scaling law by decomposing the phenomenon into token-level dynamics. While training an OLMo language model, the authors discovered that the token-level trajectories of cross-entropy are surprisingly noisy. Most are approximated by “token-wise” power laws with heterogeneous parameters that do not directly average into the aggregate power law. Additionally, they found that this “microscopic” diversity sharply narrows down to the macroscopic power law once the coverage surpasses a certain critical mass of tokens. They corroborate this argument by reproducing similar results on an artificial dataset with known ground-truth next-token distributions, verifying that similar microscopic diversity exists even in the true cross-entropy. This paper highlights the discrepancy between the macroscopic power law and noisy microscopic trajectories, illustrating how averaging different token-wise power laws induces the canonical aggregate power law.

**Strengths:**

1. It provides a novel perspective on understanding the neural scaling law by investigating token-wise microscopic behavior.
2. The authors support their arguments with experiments based on synthetic trigram models.

**Suggestions:**

1. **Any mechanistic models for this phenomenon?**: While this paper reports an interesting viewpoint for understanding the neural scaling law, it does not suggest a theoretical or mechanistic explanation for the microscopic heterogeneous scaling law or its aggregation as coverage increases. It would be insightful to introduce mechanistic models to explain this token-wise scaling law.

2. **More discussion on previous works**: The Quantization Model in [1] explains the neural scaling law through quanta, suggesting that macroscopic scaling can be understood as the learning of a sequence of quantized skills. This view has also been verified theoretically in a toy multitask sparse parity problem [2]. Although this paper mentions [1], it lacks a comprehensive discussion. This paper investigates the next-token prediction task at the token level, finding that token-wise trajectories are not quantized but rather follow individual scaling laws. It would be interesting to discuss further how this token-wise view of next-token prediction can be reconciled with the quantization model.

[1] Michaud et al., The quantization model of neural scaling, NeurIPS 2023

[2] Nam et al., An exactly solvable model for emergence and scaling laws in the multitask sparse parity problem, NeurIPS 2024

---

### Meta-Review · Area_Chair_hUCp · 2026-03-02

**Recommendation:** Accept

**Metareview:**

This work empirically studies the scaling of the token-level loss in LLMs, showing that it exhibits heterogeneous behaviour across tokens and that overall power-law scaling emerges only after averaging across tokens. The experiments are performed on both text and a synthetic trigram model. The findings are interesting and feed into a timely discussion, although the lack of a theoretical analysis/mechanistic understanding of the phenomena limits the potential impact of the work.

---

### Decision · Program_Chairs · 2026-03-02

Accept